# The Protective Role of KANK1 in Podocyte Injury

**DOI:** 10.3390/ijms25115808

**Published:** 2024-05-27

**Authors:** Keiko Oda, Kan Katayama, Liqing Zang, Masaaki Toda, Akiko Tanoue, Ryosuke Saiki, Taro Yasuma, Corina N. D’Alessandro-Gabazza, Yasuhito Shimada, Mutsuki Mori, Yasuo Suzuki, Tomohiro Murata, Toshinori Hirai, Karl Tryggvason, Esteban C. Gabazza, Kaoru Dohi

**Affiliations:** 1Department of Cardiology and Nephrology, Mie University Graduate School of Medicine, Tsu 514-8507, Mie, Japan; k-oda@med.mie-u.ac.jp (K.O.); 317ds03@m.mie-u.ac.jp (A.T.); ryosuke-s@med.mie-u.ac.jp (R.S.); mutsuki-m@med.mie-u.ac.jp (M.M.); suzuki-y@med.mie-u.ac.jp (Y.S.); tmhr0421@med.mie-u.ac.jp (T.M.); dohik@med.mie-u.ac.jp (K.D.); 2Department of Medical Biochemistry and Biophysics, Karolinska Institute, 171 77 Stockholm, Sweden; karl.tryggvason@duke-nus.edu.sg; 3Graduate School of Regional Innovation Studies, Mie University, Tsu 514-8507, Mie, Japan; liqing@doc.medic.mie-u.ac.jp; 4Department of Immunology, Mie University Graduate School of Medicine, Tsu 514-8507, Mie, Japan; t-masa@med.mie-u.ac.jp (M.T.); t-yasuma0630@med.mie-u.ac.jp (T.Y.); dalessac@clin.medic.mie-u.ac.jp (C.N.D.-G.); gabazza@med.mie-u.ac.jp (E.C.G.); 5Department of Integrative Pharmacology, Mie University Graduate School of Medicine, Tsu 514-8507, Mie, Japan; shimada.yasuhito@mie-u.ac.jp; 6Department of Pharmacy, Faculty of Medicine, Mie University Hospital, Tsu 514-8507, Mie, Japan; t.hirai@tmd.ac.jp

**Keywords:** adriamycin, albuminuria, apoptosis, KANK1, nephrotic syndrome, podocyte

## Abstract

Approximately 30% of steroid-resistant nephrotic syndromes are attributed to monogenic disorders that involve 27 genes. Mutations in *KANK* family members have also been linked to nephrotic syndrome; however, the precise mechanism remains elusive. To investigate this, podocyte-specific *Kank1* knockout mice were generated to examine phenotypic changes. In the initial assessment under normal conditions, *Kank1* knockout mice showed no significant differences in the urinary albumin-creatinine ratio, blood urea nitrogen, serum creatinine levels, or histological features compared to controls. However, following kidney injury with adriamycin, podocyte-specific *Kank1* knockout mice exhibited a significantly higher albumin-creatinine ratio and a significantly greater sclerotic index than control mice. Electron microscopy revealed more extensive foot process effacement in the knockout mice than in control mice. In addition, *KANK1*-deficient human podocytes showed increased detachment and apoptosis following adriamycin exposure. These findings suggest that KANK1 may play a protective role in mitigating podocyte damage under pathological conditions.

## 1. Introduction

Nephrotic syndrome (NS) is characterized by massive proteinuria, hypoalbuminemia, and systemic edema [1] and is categorized into steroid-sensitive NS (SSNS) and steroid-resistant NS (SRNS). Notably, approximately 50% of SRNS cases progress to end-stage renal disease within 15 years [2]. Clinically, SRNS often presents as focal segmental glomerulosclerosis (FSGS), with genetic factors implicated in 5–10% of adult cases. [3]. Recent studies have identified that nearly 30% of SRNS cases result from mutations in 1 of 27 specific genes [4]. The proteins encoded by 27 genes act as functional proteins (e.g., glomerular slit membrane components, laminin/integrin signaling components, actin-binding proteins, actin-regulating small GTPases, lysosomal proteins, transcription factors, and proteins of coenzyme Q10 biosynthesis) [5]. With advancements in next-generation sequencing (NGS), the discovery of additional related genes is anticipated. For instance, *KANK1*, *KANK2*, and *KANK4* have been associated with NS [6], despite not being among the 27 previously identified genes [4]. KANK family proteins play a role in actin regulation, which might be similar to ARHGDIA, the mutation of which is reported to cause SRNS [5].

*KANK1*, known as the KN motif and ankyrin repeat domain 1, has been recognized as a tumor suppressor gene in renal cell carcinoma [7]. The KANK family, which encompasses KANK1 through KANK4, is characterized by the presence of KN motifs, coiled-coil domains, and ankyrin repeat domains (ANKRDs) [8,9]. KANK1 is predominantly cytoplasmic and plays a pivotal role in the organization of the actin cytoskeleton [10], cell polarity [11], and focal adhesions [12,13,14]. KANK1 exists in two isoforms; the longer variant is tissue-specific, with predominant expression in the heart and kidneys [15]. KANK1 has been identified as 1 of the 677 genes significantly enriched in human glomeruli [16].

Recent research has linked the *KANK1* gene with nephrotic syndrome [6]. Although *Kank1* RNAi mice exhibited a tendency towards proteinuria, the results were not statistically significant [17]. Given the conservation of the gene between humans and zebrafish, a *kank1* knockout zebrafish model was developed, featuring a C-to-T point mutation in exon 3 of the *kank1* gene. The dye filtration assay in zebrafish has proven to be an effective tool for examining glomerular permeability [18,19]. In preliminary experiments using this model, 500 kDa fluorescein isothiocyanate (FITC)- and 10 kDa rhodamine-conjugated dextran were injected into *kank1* knockout or wild-type (WT) zebrafish embryos. The *kank1* knockout group displayed an uptake of 500 kDa dextran in the proximal tubule, unlike the WT group. Based on this observation, we hypothesized that podocyte-specific *Kank1* inactivation in mice might manifest as a kidney phenotype.

## 2. Results

### 2.1. KANK1 Protein Is Expressed in Both Human and Mouse Podocytes

Co-immunofluorescence studies on frozen human kidney sections revealed that KANK1 was localized in the glomeruli, exhibiting co-localization with the podocyte-specific protein NPHS1 (Figure 1a). In addition, KANK1 expression was observed in the tubular basement membranes, albeit with minimal co-localization with SGLT2, a protein present in the brush borders of the proximal tubules (Appendix A). In mouse models, KANK1 expression was similarly observed in the glomeruli, where it co-localized with the podocyte-specific protein SYNPO (Figure 1b).

### 2.2. Kank1 Was Successfully Knocked Out in Mouse Podocytes

*Kank1* was engineered to contain loxP sites in the 5th and 9th introns of *Kank1* (Figure 2a). To specifically target and remove exons 6–9 in podocytes, we used mice carrying the podocin-Cre (pod-Cre^Tg/+^) gene. By crossing *Kank1* flox/flox (*Kank1*^fl/fl^) mice with pod-Cre^Tg/+^ mice, we generated offspring using various combinations of these genes, including *Kank1*^fl/fl^, *Kank1*^fl/fl^ pod-Cre^Tg/+^, and *Kank1*^+/+^ pod-Cre^Tg/+^ mice. The resulting genotypes were discerned by distinct band patterns during genotyping: *Kank1*^+/+^pod-Cre^Tg/+^ mice exhibited a 459-bp band, while *Kank1*^fl/fl^ and *Kank1*^fl/fl^ pod-Cre^Tg/+^ mice displayed a 587-bp band (Figure 2b). Notably, a 102-bp band was present in *Kank1*^+/+^ pod-Cre^Tg/+^ and *Kank1*^fl/fl^ pod-Cre^Tg/+^ mice but absent in *Kank1*^fl/fl^ mice (Figure 2b). Subsequent real-time polymerase chain reaction (PCR) revealed a marked reduction in *Kank1* mRNA expression in the glomeruli of *Kank1*^fl/fl^ pod-Cre^Tg/+^ mice compared to their *Kank1*^fl/fl^ counterparts (0.14 ± 0.05 vs. 1.28 ± 0.26, *p* = 0.0001, Figure 2c). Western blotting corroborated these findings, showing undetectable KANK1 protein levels in the *Kank1*^fl/fl^ pod-Cre^Tg/+^ group (Figure 2d). Appendix A shows full blots. Immunofluorescence of frozen mouse kidney sections confirmed the diminished expression of KANK1 in the glomeruli of *Kank1*^fl/fl^ pod-Cre^Tg/+^ mice, in stark contrast to the clear expression observed in the *Kank1*^+/+^ pod-Cre^Tg/+^ and *Kank1*^fl/fl^ groups (Figure 2e).

### 2.3. The Inactivation of Kank1 in Podocytes at Two or Six Months Did Not Result in Albuminuria

At two months, the urinary albumin-creatinine ratio (ACR) was consistent across the *Kank1*^+/+^pod-Cre^Tg/+^, *Kank1*^fl/fl,^ and *Kank1*^fl/fl^ pod-Cre^Tg/+^ groups (Figure 3a). Similarly, no notable differences were observed in the blood urea nitrogen (BUN) or serum creatinine (Cr) levels among the groups (Figure 3b). However, the sclerotic index was significantly higher in the *Kank1*^fl/fl^ pod-Cre^Tg/+^ group than in the *Kank1*^fl/fl^ group (0.59 ± 0.15 vs. 0.34 ± 0.08, *p* = 0.0201), with no marked difference when compared to the *Kank1*^+/+^ pod-Cre^Tg/+^ group (Figure 3c). The fibrotic index showed no significant differences among the three groups (Figure 3c). In addition, an electron microscopic analysis revealed no significant disparities between the groups at this age (Figure 3d).

At six months, the urinary ACR remained stable in the *Kank1*^fl/fl^ pod-Cre^Tg/+^ group compared to that in the *Kank1*^fl/fl^ group (Appendix A). Similarly, there was no significant difference in BUN or Cr levels between *Kank1*^fl/fl^ and *Kank1*^fl/fl^ pod-Cre^Tg/+^ mice at six months (Appendix A). However, both sclerotic (0.86 ± 0.41 vs. 0.31 ± 0.25, *p* = 0.0181) and fibrotic (0.19 ± 0.11 vs. 0.08 ± 0.03, *p* = 0.0433) indices were significantly higher in the *Kank1*^fl/fl^ pod-Cre^Tg/+^ group than in the *Kank1*^fl/fl^ group (Appendix A).

### 2.4. The Inactivation of Kank1 in Podocytes at One Year Did Not Result in Abnormalities in Urine, Blood, or Histological Tests

At one year, the urinary ACR showed no significant difference between the *Kank1*^fl/fl^ and *Kank1*^fl/fl^ pod-Cre^Tg/+^ groups (Figure 4a). Similarly, BUN and Cr levels were comparable between the two groups at this age (Figure 4b). There was no significant difference in sclerotic or fibrotic indices between the two groups (Figure 4c). Furthermore, electron microscopic analysis revealed no notable disparities in the results between the groups (Figure 4d).

### 2.5. Compensation by other Members of the Kank Family Was Not Evident in Podocyte-Specific Kank1 Knockout Mice

One possible explanation for the observed outcomes under normal conditions is that compensatory mechanisms from other members of the Kank family, specifically Kank2, Kank3, and Kank4, may mitigate phenotypic effects in the Kank1^fl/fl^ pod-Cre^Tg/+^ group. However, real-time PCR analyses conducted on the glomeruli of Kank1^fl/fl^ pod-Cre^Tg/+^ mice yielded results consistent with those obtained from Kank1^fl/fl^ mice (Figure 5a–c).

### 2.6. Albuminuria and Glomerular Sclerosis Evident in Podocyte-Specific Kank1 Knockout Mice under Pathological Conditions

To investigate potential phenotypic differences under disease conditions, we induced kidney injury in adriamycin-treated mice. Following a two-week period post-administration, we observed a significant increase in the urinary ACR in the Kank1^fl/fl^ pod-Cre^Tg/+^ group compared to the Kank1^fl/fl^ group. The logarithmic urinary ACR was also significantly higher in the Kank1^fl/fl^ pod-Cre^Tg/+^ group than in the Kank1^fl/fl^ group (7.4 ± 1.1 vs. 5.5 ± 0.8, *p* = 0.0157; Figure 6a). No significant differences were detected in BUN or Cr levels between the two groups (Figure 6b). Post-adriamycin treatment, the sclerotic index was significantly elevated in the Kank1^fl/fl^ pod-Cre^Tg/+^ group compared to the Kank1^fl/fl^ group (0.46 ± 0.09 vs. 0.29 ± 0.04, *p* = 0.0056), although the fibrotic index showed no significant difference (Figure 6c). An electron microscopic analysis revealed a marked decrease in the density of foot processes per micrometer of glomerular basement membranes in the Kank1^fl/fl^ pod-Cre^Tg/+^ group relative to the Kank1^fl/fl^ group (1.00 ± 0.18 vs. 1.50 ± 0.15, *p* = 0.0051; Figure 6d).

### 2.7. Generation of the Human Immortalized KANK1 Knockout (KANK1KO) Podocyte Cell Line

The observation of a link between Kank1 deficiency and kidney dysfunction in a mouse model prompted us to establish an in vitro human KANK1KO podocyte model. We investigated whether or not similar effects occurred in KANK1-deficient human podocytes. Rhodamine–phalloidin staining revealed KANK1 protein localization in the podocyte cytoplasm, specifically at the actin fiber edges (Figure 7a). Despite the nonspecific presence of nuclear stain, even without the primary KANK1 antibody, CRISPR-Cas9 technology facilitated the development of a KANK1KO podocyte cell line. This revealed a homozygous insertion (c.110 ins A) in exon 2 of KANK1, leading to a frameshift, premature stop codon, and truncated p.Y37* protein (Figure 7b). In KANK1KO podocytes, KANK1 protein expression at approximately 180 kDa was undetectable, unlike the visible band in WT podocytes (Figure 7c). Appendix A shows the full blot. Following adriamycin treatment, KANK1KO podocytes exhibited a significant reduction in cell attachment and an increase in apoptosis compared to WT podocytes (Figure 7d,e), emphasizing the protective role of the gene against podocyte injury.

## 3. Discussion

In this study, we successfully demonstrated the suppression of the *Kank1* gene in podocytes at both the mRNA and protein levels using podocyte-specific *Kank1* knockout mice. The KANK1 protein, which co-localizes with podocyte-specific markers in humans, was confirmed to be expressed in podocytes. Its localization coincided with that of actin fiber markers, consistent with previous findings of its association with the actin cytoskeleton [9]. Under normal conditions, no notable phenotypic alterations were detected in podocyte-specific *Kank1* knockout mice for up to one year, which is consistent with another report of the *Kank1* knockout mouse model [20]. Compensation by other members of the *Kank* family was not evident in the present study. However, following adriamycin-induced renal injury, the mice manifested a pronounced elevation in the urinary ACR and sclerotic index relative to controls. In addition, human KANK1KO podocytes displayed an increased propensity for detachment and apoptosis following adriamycin exposure. Overall, these findings highlight KANK1’s potential as a protective factor against podocyte injury in disease states.

In the mouse models developed in this study, the omission of exon 6–9 in *Kank1* led to a frameshift mutation, resulting in a premature stop codon. Consequently, the KANK1 protein was truncated, reducing its length from 1360 to 1052 amino acids and lacking 5 ANKRDs. Recent studies have elucidated the function of ANKRDs [21], and loss of domain function can lead to the results in the present study. Previous research has indicated that a singular rare variant in *KANK1* and three in *KANK2* were associated with minimal change nephrotic syndrome (MCNS), while two rare variants in *KANK4* were linked to FSGS [6], aligning with the mild phenotype observed in podocyte-specific *Kank1* knockout mice. In addition, a study on *Kank1* RNAi mice reported no significant increase in proteinuria [17], corroborating our findings. Although KANK1 may not be a direct causative factor, the results of the present study suggest that KANK1 can contribute to albuminuria.

In light of previous research suggesting that adriamycin exposure results in certain phenotypes [22], we administered this compound to both *Kank1*^fl/fl^ pod-Cre^Tg/+^ and *Kank1*^fl/fl^ mice. Following adriamycin-induced injury, *Kank1*^fl/fl^ pod-Cre^Tg/+^ mice exhibited a notable increase in the urinary ACR and sclerotic index relative to *Kank1*^fl/fl^ mice. Considering that elevated serum hemopexin levels have been implicated as a permeability factor in patients with MCNS [23,24,25], we assessed these levels in both mouse models before and after adriamycin administration. The results indicated a significant elevation in serum hemopexin levels in *Kank1*^fl/fl^ pod-Cre^Tg/+^ mice post-treatment, whereas no such increase was observed in *Kank1*^fl/fl^ mice (Appendix A).

There are several limitations associated with the current study. First, we did not perform experiments involving alternative inducible mouse models other than the Adriamycin model. Second, the gene expression analyses were not performed in *Kank1* knockout mice or control mice in a healthy state or during renal damage. Third, other urinary biomarkers were not tested in the current mouse model. Fourth, the effects of drugs used for nephrotic syndrome, such as angiotensin-converting enzyme inhibitors, angiotensin receptor blockers, sodium-glucose cotransporter-2 inhibitors, and steroids, were not tested in the current mouse model. Fifth, only one KANK1KO podocyte cell line was generated; however, the knockout of *KANK1* was observed at the genomic and protein levels, while artifacts in other genes could not be excluded. Finally, primary glomerular culture could not be performed due to technical problems.

In summary, our study revealed that podocyte-specific *Kank1* knockout mice, following adriamycin-induced kidney injury, display a significant rise in ACR, an augmented sclerotic index, and enhanced foot process effacement compared to their control counterparts. Furthermore, KANK1KO podocytes exhibited an increased susceptibility to detachment after adriamycin treatment, correlating with a greater propensity for apoptosis. Taken together, these results suggest that KANK1 may play a protective role in mitigating podocyte damage under pathological conditions. Therapeutic strategies for KANK1 gene mutations have not yet been established, and further research is required. As it is predicted that those with KANK1 gene mutations are more likely to develop kidney diseases, it is important for people with KANK1 gene mutations to be aware of the risk of developing kidney diseases and to strive to protect their kidneys.

## 4. Materials and Methods

### 4.1. Mice

C57BL/6 *Kank1*^fl/fl^ mice were generated by inserting loxP sites in introns 5 and 9 of the *Kank1* gene using Cyagen (Santa Clara, CA, USA). C57BL/6 pod-Cre^Tg/+^ mice were purchased from the Jackson Laboratory (Sacramento, CA, USA). All experimental protocols were approved by the Animal Care and Use Committee of Mie University (No. 27-32), and all experiments were performed in accordance with approved guidelines.

Genomic DNA was extracted from the tails of *Kank1*^+/+^pod-Cre^Tg/+^, *Kank1*^fl/fl^, and *Kank1*^fl/fl^ pod-Cre^Tg/+^ mice using a FavorPrep Tissue Genomic DNA Extraction Mini Kit (FAVORGEN, Ping-Tung, Taiwan). Polymerase chain reaction (PCR) amplification was carried out with Hotstartaq (QIAGEN, Hilden, Germany) under the following conditions: 96 °C for 15 min; 30 cycles of 96 °C for 45 s, 57 °C for 45 s, and 72 °C for 1 min; and extension at 72 °C for 15 min. The PCR primers used were as follows: Kank1_f2, 5′-TCATTTCTTGCATAGCCGGTA-3′ and Kank1_r2, 5′-CTGTGCTATTCCTGACCAGT-3′ for Kank1; and oIMR1084, 5′-GCGGTCTGGCAGTAAAAACTATC-3′ and oIMR1085, 5′-GTGAAACAGCATTGCTGTCACTT-3′. The PCR products were analyzed by gel electrophoresis on a 1.5% agarose gel. Adriamycin at a dose of 15 mg/kg body weight was injected via the mouse tail vein [26] once, and the mice were sacrificed after two weeks.

### 4.2. Isolation of Mouse Glomeruli

Glomeruli were isolated from *Kank1*^fl/fl^ and *Kank1*^fl/fl^ pod-Cre^Tg/+^ mice at six months by a perfusion method involving magnetic beads with collagenase digestion for RNA and without collagenase digestion for protein experiments [27].

### 4.3. RNA Extraction and Reverse Transcriptase (RT)-PCR

Total RNA was extracted from the glomeruli of the *Kank1*^fl/fl^ or *Kank1*^fl/fl^ pod-Cre^Tg/+^ group (*n* = 4 each) using the TRIzol Plus RNA purification kit (ThermoFisher Scientific, Waltham, MA, USA), and cDNA was synthesized using SuperScript III reverse transcriptase (ThermoFisher Scientific). Real-time RT-PCR was performed using a StepOnePlus real-time PCR system (ThermoFisher Scientific). The oligonucleotide primers used were as follows: Kank1, 5′-TGTCAGCCTGCAACTTACTGA-3′ (forward) and 5′-CTGACTGGACACTCGGAACC-3′ (reverse); Kank2, 5′- GCACACAGCCAAGAAGATCA-3′ (forward) and 5′- CAGGGTTCTCAGGCTGTACC-3′ (reverse); Kank3, 5′-TTTCCAGCCTGCTACTGGAT-3′ (forward) and 5′-GCCATGTCTTCCTCCTCTTG-3′ (reverse); Kank4, 5′-TCGCCATTGTCAAGCTACTG-3′ (forward) and 5′-TTTCCAGACAACAGCCATGT-3′ (reverse); GAPDH, 5′-CGTCCCGTAGACAAAATGGT-3′ (forward) and 5′-GAATTTGCCGTGAGTGGAGT-3′ (reverse).

### 4.4. Immunofluorescence for KANK1 in Humans or WT Mice

Frozen sections of normal human kidneys were purchased from Zyagen (San Diego, CA, USA) and fixed in acetone at −20 °C for 10 min. Blocking was performed with 10% normal goat serum in phosphate buffer saline (PBS) for 30 min at room temperature. The sections were incubated with rabbit anti-KANK1 (1:100) (Proteintech, Rosemont, IL, USA) and mouse anti-NPHS1 (1:400) (50A9) [28] or rabbit anti-KANK1 (1:100) and mouse anti-SGLT2 (1:100) (Santa Cruz Biotechnology, Santa Cruz, CA, USA) in 10% normal goat serum in PBS at 4 °C overnight. After washing with PBS three times, the sections were incubated with goat anti-rabbit Alexa Fluor 488 (1:400) (ThermoFisher Scientific) and goat anti-mouse Alexa Fluor 568 (1:400) (ThermoFisher Scientific) in 1% bovine serum albumin (BSA) in PBS for 1 h. The sections were mounted after staining with 4′,6-diamindino-2-phenylindole (DAPI) (Invitrogen, Carlsbad, CA, USA) and examined using an FV1000 confocal laser scanning microscope (Olympus, Tokyo, Japan).

Frozen sections of WT mouse kidneys were fixed in acetone in a similar manner. After blocking with 10% normal goat serum in PBS, the sections were incubated with rabbit anti-KANK1 (1:100) (Proteintech) and guinea pig anti-SYNPO (1:500) (PROGEN, Heidelberg, Germany) with 10% normal goat serum in PBS at 4 °C overnight. Goat anti-rabbit Alexa Fluor 488 (1:400) (ThermoFisher Scientific) and goat anti-guinea pig Alexa Fluor 568 (1:400) (ThermoFisher Scientific) were used as secondary antibodies.

### 4.5. Western Blotting

Glomeruli were isolated from six-month-old *Kank1*^fl/fl^ or *Kank1*^fl/fl^ pod-Cre^Tg/+^ mice without collagenase digestion. The samples were separated on NuPAGE 4–12% Bis-Tris gels (ThermoFisher Scientific) and transferred to polyvinylidene difluoride membranes. After blocking with Tris-buffered saline containing 0.1% Tween 20 and 5% milk, the upper membrane was incubated with anti-KANK1 antibody (1:1000) (ATLAS ANTIBODIES, Bromma, Sweden) at 4 °C overnight. The membrane was then incubated for 1 h at room temperature with horseradish peroxidase (HRP)-labeled donkey anti-rabbit antibody (1:3000) (GE Healthcare, Chicago, IL, USA) and developed using Amersham ECL Prime (GE Healthcare) according to the manufacturer’s instructions. The lower membrane (<100 kDa) was incubated with anti-β-actin antibody (1:2000) (Cell Signaling, Danvers, MA, USA) at 4 °C overnight, incubated for 1 h at room temperature with HRP-labeled sheep anti-mouse antibody, and developed in the same way.

### 4.6. Immunofluorescence Studies in Mice

Snap-frozen kidney blocks from two-month-old animals were prepared in Tissue-Tek OCT compound (Sakura Finetek, Tokyo, Japan) and sliced into 10 µm sections, and thereafter, they were made on silane-coated microslides (MUTO PURE CHEMICALS, Tokyo, Japan). The sections were fixed in acetone for 10 min at −20 °C. After blocking with 10% normal goat serum in PBS for 60 min, the sections were incubated with anti-KANK1 antibody (1:500; ATLAS ANTIBODIES) or no primary antibody overnight at 4 °C. After washing with PBS, the sections were incubated with goat anti-rabbitAlexaFluor488 (1:500) at room temperature for 1 h and DAPI (1:3000) for 5 min and then examined using an FV1000 confocal laser scanning microscope (Olympus, Tokyo, Japan).

### 4.7. Urine and Blood Analyses

Urinary ACR, BUN, and Cr were measured in the *Kank1*^+/+^ pod-Cre^Tg/+^, *Kank1*^fl/fl^, and *Kank1*^fl/fl^ pod-Cre^Tg/+^ groups at 2 months (*n* = 6) and the *Kank1*^fl/fl^ group, and the *Kank1*^fl/fl^ pod-Cre^Tg/+^ group at 6 months or 1 year (*n* = 6 each). The urinary ACR was measured using an immunoturbidimetric method for urinary albumin (Shibayagi, Gunma, Japan) and an enzymatic assay for urinary creatinine (Wako, Tokyo, Japan). BUN and Cr levels were measured as previously described [29].

### 4.8. Light Microscopic Analyses

For periodic acid-Schiff (PAS) and Masson trichrome (MT) staining, six kidneys from three groups (*Kank1*^+/+^ pod-Cre^Tg/+^, *Kank1*^fl/fl^, and *Kank1*^fl/fl^ pod-Cre^Tg/+^) at two months were examined (*n* = 6 for each group). At six months or one year, six kidneys from two groups, *Kank1*^fl/fl^, and *Kank1*^fl/fl^ pod-Cre^Tg/+^, were examined. The sclerotic index (0 = 0%, 1 = 1–25%, 2 = 26–50%, 3 = 51–75%, 4 = 76–100%) was evaluated by examining 30 glomeruli in PAS-stained specimens from each mouse. The fibrotic index (0 = 0%, 1 = 1–25%, 2 = 26–50%, 3 = 51–75%, 4 = 76–100%) was evaluated by examining 20 areas in the MT-stained specimen of each mouse [30].

### 4.9. Transmission Electron Microscopic Analyses

Mouse kidneys were cut into 1–2 mm blocks, pre-fixed in 2.5% glutaraldehyde and 2% PFA in 0.1 M phosphate buffer (pH 7.4), and post-fixed in 2% osmium tetra-oxide for 2 h at 4 °C. The samples were dehydrated in graded ethanol and embedded in epoxy resin. Ultrathin sections stained with uranyl acetate for 10 min and lead staining solution for 5 min were examined using a HITACHI H-700 (Hitachi, Tokyo, Japan). To assess the extent of podocyte foot process effacement, foot processes along at least 400 µm of the total glomerular basement membrane length were quantified in 4 glomeruli of the *Kank1*^fl/fl^ or *Kank1*^fl/fl^ pod-Cre^Tg/+^ group after adriamycin treatment (each *n* = 4) using the ImageJ software (version 1.54) program (National Institutes of Health, Bethesda, MD, USA), as previously reported [31,32].

### 4.10. Immunofluorescence in Human Podocytes

Human immortalized podocytes at 37 °C [33], cultured in RPMI 1640 medium (Wako) with 10% fetal bovine serum, insulin-transferrin-selenium-A (Wako), and penicillin/streptomycin, were fixed with 4% paraformaldehyde (PFA) in PBS at room temperature for 20 min, permeabilized with 0.1% Triton X-100 in PBS, and blocked with 2% BSA in PBS for 1 h at room temperature. The cells were incubated with anti-KANK1 antibody (1:200; ATLAS ANTIBODIES) or without the primary antibody at 4 °C overnight. After washing with PBS three times, the cells were incubated with goat anti-rabbitAlexaFluor488 (1:500) and rhodamine-phalloidin (1:500) (Invitrogen, Carlsbad, CA, USA) at room temperature for 1 h, followed by DAPI (1:3000) at room temperature for 5 min. The results were examined using an FV1000 confocal laser-scanning microscope (Olympus).

### 4.11. Generation of a KANK1KO Human Immortalized Podocyte Cell Line

Two different guide RNAs, *KANK1*, gRNA1, and gRNA2, which target exon 2 of *KANK1*, were tested. The sequences of *KANK1* gRNA1 and gRNA2 were 5′-GTCTAGTTGATAACCATAGG-3′ and 5′-TGACACCGGGTGAGTTCAGA-3′, respectively. The efficiency of genome editing using the GeneArt Genomic Cleavage Detection Kit (ThermoFisher Scientific) was higher in *KANK1* gRNA1 than in *KANK1* gRNA2; therefore, *KANK1* gRNA1 was used. 0.5 × 10^5^ cells were transfected with Lipofectamine CRISPRMAX (ThermoFisher Scientific) 1.5 µL, Cas9 nuclease (1.5 µg), Cas9 plus reagent (2.5 µL), and *KANK1* gRNA1 1.25 µg and incubated at 33 °C for 3 days. Single-cell cloning was performed by serial dilutions in 96-well plates. Genomic DNA was extracted from each clone, and a sequence analysis was performed to detect indels.

### 4.12. Detachment Assays

WT and KANK1KO podocyte cell lines were separately cultured in 24-well plates. The number of cells per field was counted to establish a baseline number. The cells were treated with 0.5 µg/mL adriamycin for a total of 48 h. The detachment assay was modified according to a previous study [34]. Specifically, the number of detached cells in 5 wells was evaluated at pre-treatment, 24 h after treatment, and 48 h after treatment in 3 independent experiments.

### 4.13. Apoptosis Assays

WT or KANK1KO podocyte cell lines were treated with 0.5 µg/mL adriamycin for 24 h, and the cells were harvested for an Annexin V-FITC/propidium iodide (PI) flow cytometry apoptosis assay to examine the effects of adriamycin.

### 4.14. Statistical Analyses

Data are expressed as the mean ± standard deviation (SD). Statistical analyses were performed using the StatView software package (version 5.0, SAS statistical software program, SAS Institute, Cary, NC, USA). A one-way analysis of variance (ANOVA) was used to analyze the data among the three mouse groups, followed by the post-hoc Scheffe test. Student’s *t*-test was used to compare the data between the two mouse groups. *p* < 0.05 was considered significant.

## Figures and Tables

**Figure 1 ijms-25-05808-f001:**
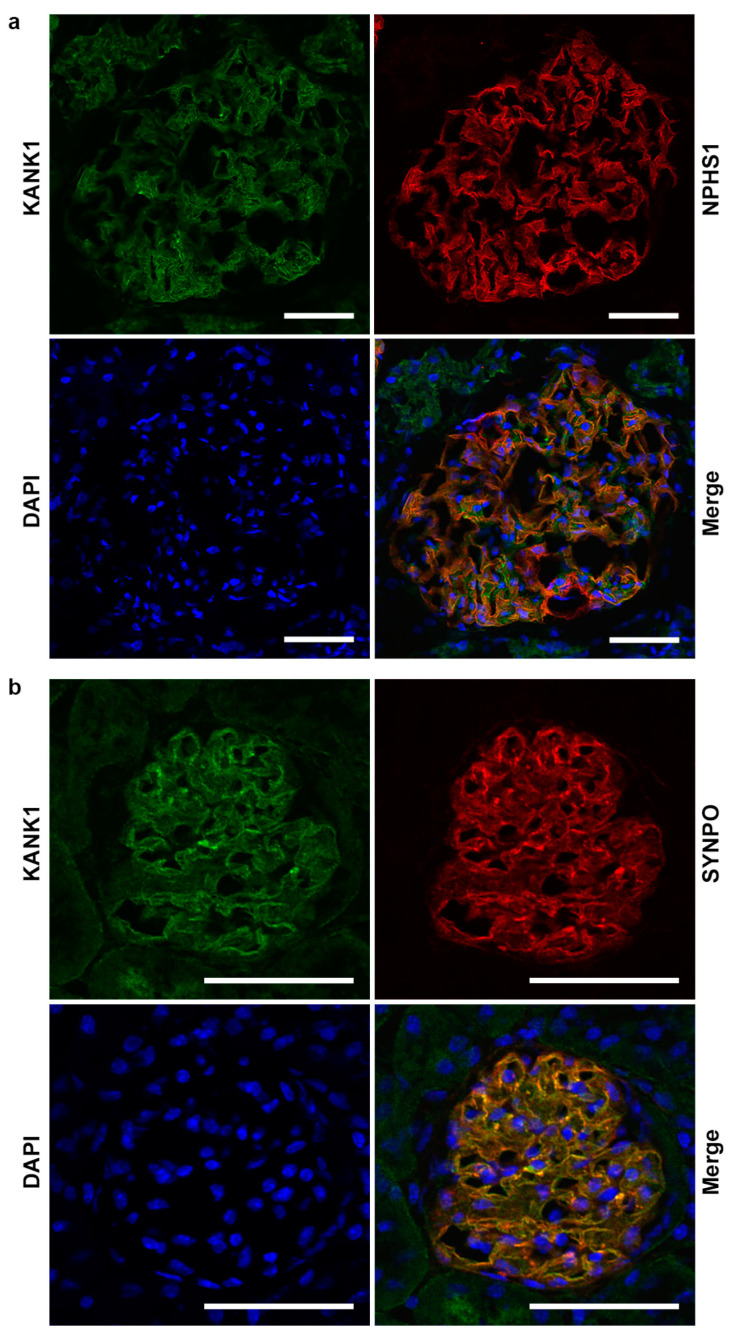
Expression of KANK1 in podocytes from human and mouse kidneys. (**a**) In human glomeruli, KANK1 showed extensive localization with NPHS1, a marker specific to podocytes. Scale bars indicate 50 µm. (**b**) In mouse glomeruli, KANK1 exhibited extensive localization with SYNPO, a marker specific to podocytes. DAPI, 4′,6-diamidino-2-phenylindole. Scale bars indicate 50 µm.

**Figure 2 ijms-25-05808-f002:**
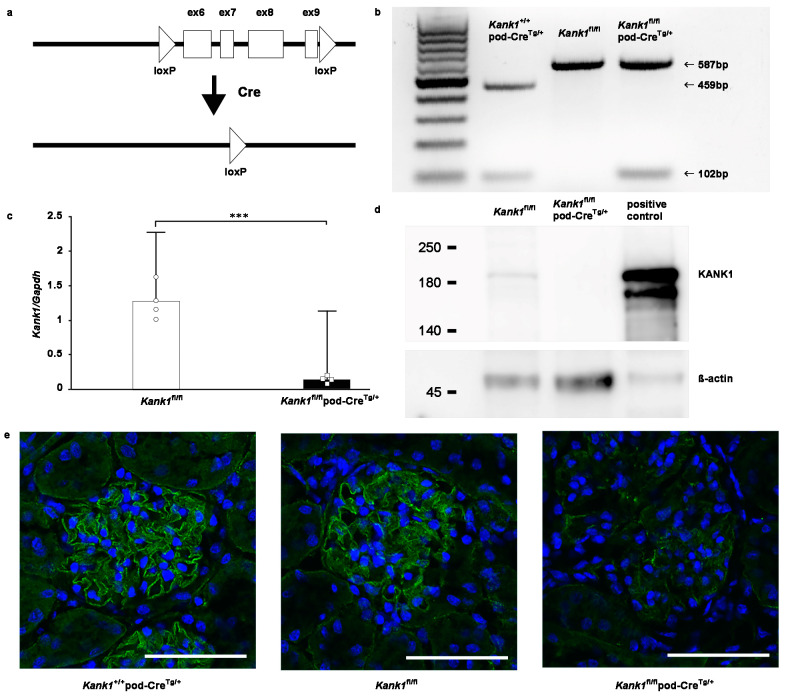
Successful knockout of podocyte-specific *Kank1*. (**a**) LoxP sites were strategically inserted into introns 5 and 9 of the *Kank1* gene. (**b**) Genotyping revealed a 459-bp band in *Kank1*^+/+^pod-Cre^Tg/+^ mice, whereas a 587-bp band was present in both *Kank1*^fl/fl^ and *Kank1*^fl/fl^pod-Cre^Tg/+^ mice using *Kank1* primers. A 102-bp band was observed in both *Kank1*^+/+^ pod-Cre^Tg/+^ and *Kank1*^fl/fl^ pod-Cre^Tg/+^ mice, but it was absent in *Kank1*^fl/fl^ mice. (**c**) *Kank1* mRNA expression was significantly reduced in the glomeruli of *Kank1*^fl/fl^ pod-Cre^Tg/+^ mice compared to *Kank1*^fl/fl^ mice at 6 months (*** *p* < 0.001). (**d**) A Western blot analysis showed that KANK1 protein expression was not detectable in the glomeruli isolated from *Kank1*^fl/fl^ pod-Cre^Tg/+^ mice, in contrast to *Kank1*^fl/fl^ mice at six months. β-actin expression was similar between the *Kank1*^fl/fl^ and *Kank1*^fl/fl^ pod-Cre^Tg/+^ groups. (**e**) At two months, KANK1 protein expression was significantly reduced in the glomeruli of *Kank1*^fl/fl^ pod-Cre^Tg/+^ mice when compared to the *Kank1*^+/+^ pod-Cre^Tg/+^ group or *Kank1*^fl/fl^ group. Scale bars indicate 50 µm.

**Figure 3 ijms-25-05808-f003:**
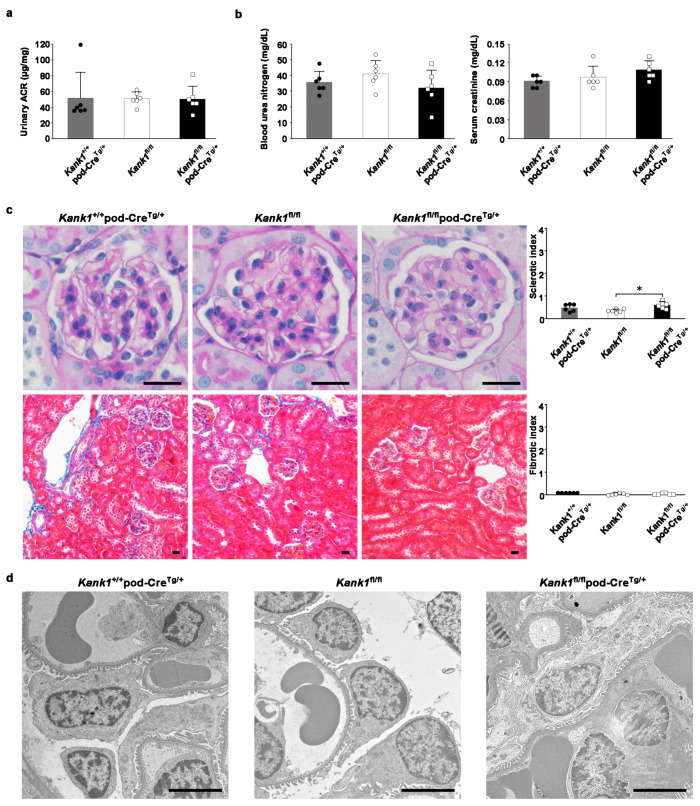
Inactivation of *Kank1* in podocytes at two months does not result in albuminuria. (**a**) The urinary albumin-creatinine ratio (ACR) was comparable across all three groups. (**b**) No significant differences were observed in blood urea nitrogen (BUN) or serum creatinine (Cr) levels among the groups. (**c**) The sclerotic index at two months was significantly higher in the *Kank1*^fl/fl^ pod-Cre^Tg/+^ group compared to the *Kank1*^fl/fl^ group (* *p* < 0.05). In contrast, the *Kank1*^+/+^ pod-Cre^Tg/+^ group did not show a significant difference from the *Kank1*^fl/fl^ pod-Cre^Tg/+^ group. In addition, the fibrotic index remained consistent among the three groups. Scale bars indicate 20 µm. (**d**) Microscopic analysis confirmed these findings, with no significant differences in the electron microscopic studies across the groups. Scale bars indicate 4 µm.

**Figure 4 ijms-25-05808-f004:**
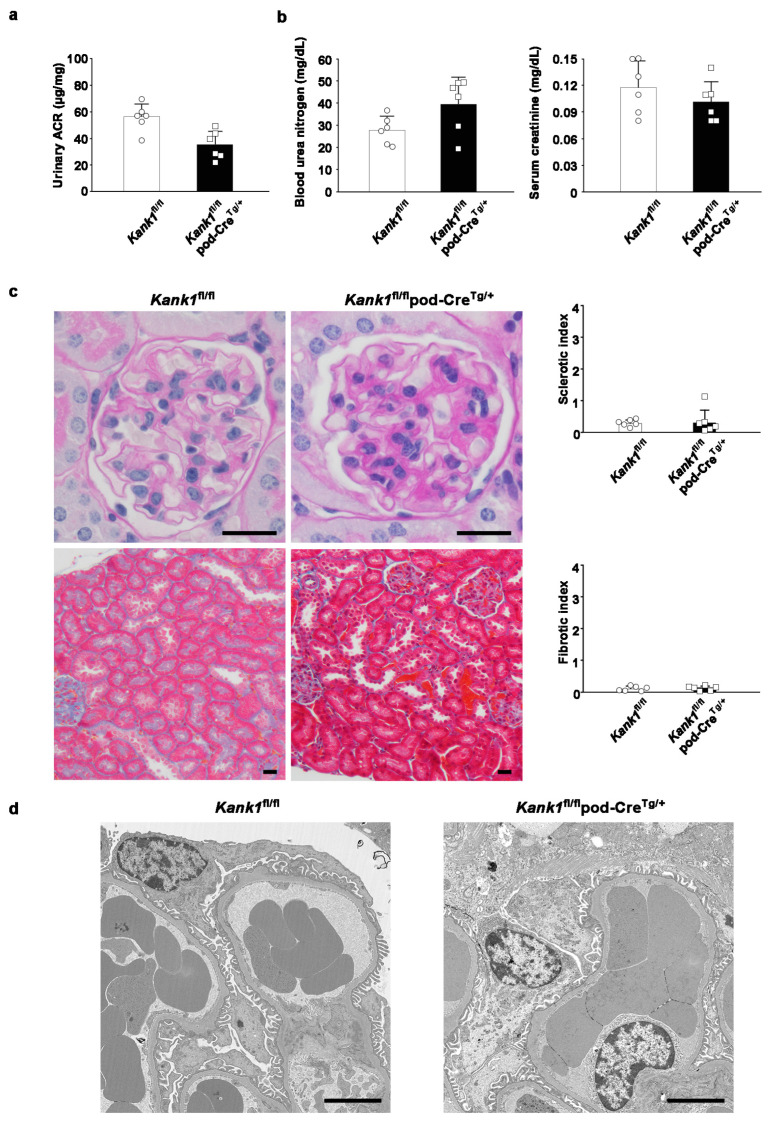
Inactivation of *Kank1* in podocytes at one year was not associated with any detectable abnormalities in urine, blood, or histological tests. (**a**) There was no significant difference in the albumin-creatinine ratio (ACR) at one year between the *Kank1*^fl/fl^ group and the *Kank1*^fl/fl^ pod-Cre^Tg/+^ group. (**b**) The blood urea nitrogen (BUN) and serum creatinine (Cr) levels were comparable between the two groups at one year. (**c**) There was no significant difference in sclerotic or fibrotic index between the two groups. Scale bars indicate 20 µm. (**d**) There was no significant difference in electron microscopic results between the two groups at one year. Scale bars indicate 4 µm.

**Figure 5 ijms-25-05808-f005:**
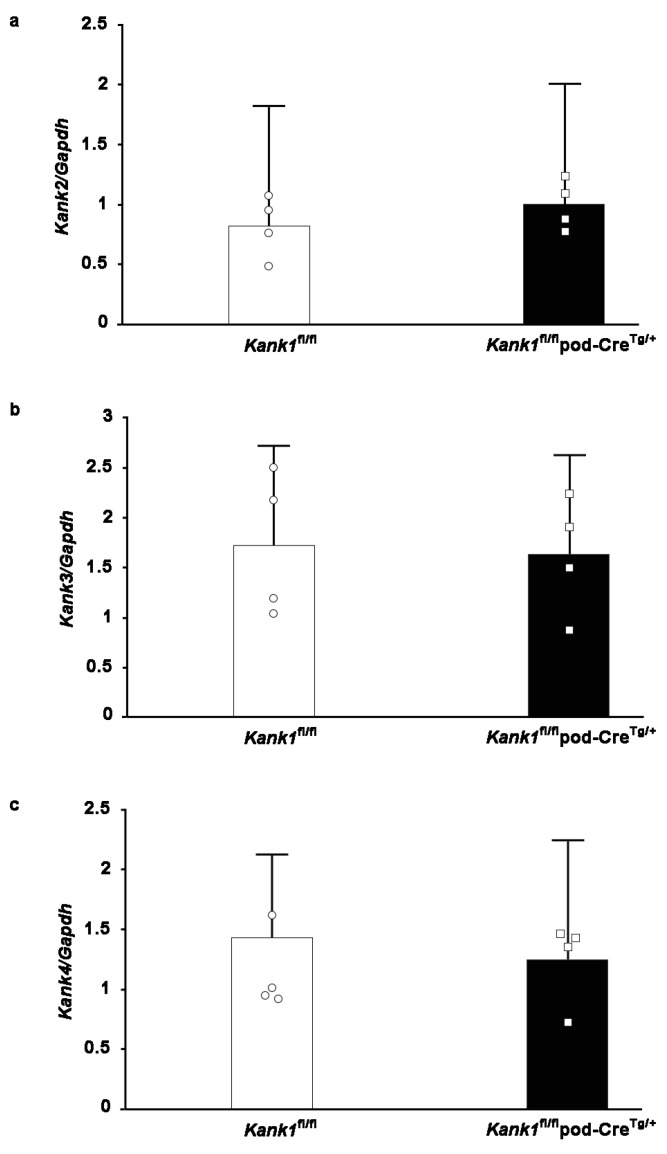
Compensation by the other Kank family members was not observed in podocyte-specific Kank1 knockout mice. (**a**–**c**) The mRNA levels of Kank2, Kank3, and Kank4 in the glomeruli of the Kank1^fl/fl^pod-Cre^Tg/+^ group showed no significant difference when compared to the Kank1^fl/fl^ group.

**Figure 6 ijms-25-05808-f006:**
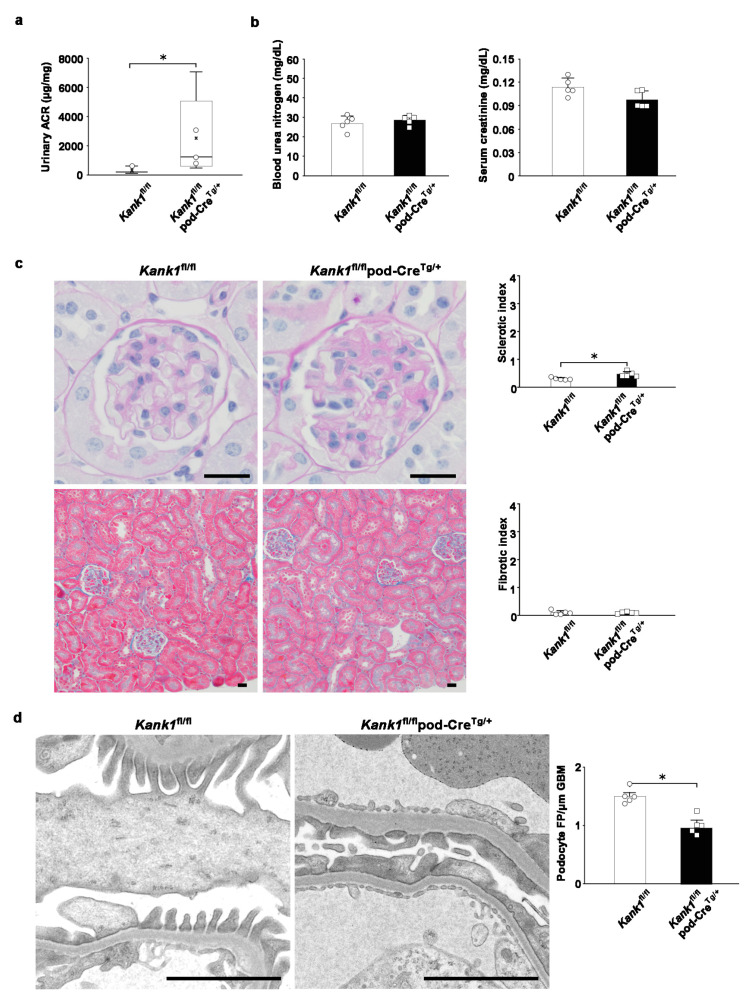
Mild glomerular sclerosis was observed in podocyte-specific Kank1 knockout mice after adriamycin-induced kidney injury. (**a**) Two weeks after adriamycin injection, the urinary albumin-creatinine ratio (ACR) was significantly increased in the Kank1^fl/fl^ pod-Cre^Tg/+^ group compared to the Kank1^fl/fl^ group (log ACR; 5.5 ± 0.8 vs. 7.4 ± 1.1, * *p* < 0.05). (**b**) There was no significant difference in BUN or Cr levels between the Kank1^fl/fl^ group and the Kank1^fl/fl^ pod-Cre^Tg/+^ group. (**c**) The sclerotic index after adriamycin injection was significantly higher in the Kank1^fl/fl^ pod-Cre^Tg/+^ group than in the Kank1^fl/fl^ group (* *p* < 0.05), while the fibrotic index was not significant between the two groups. Scale bars indicate 20 µm. (**d**) Foot processes (FP) per micrometer of glomerular basement membranes (GBM) were significantly lower in the Kank1^fl/fl^ pod-Cre^Tg/+^ group than in the Kank1^fl/fl^ group (* *p* < 0.05). Scale bars indicate 2 µm.

**Figure 7 ijms-25-05808-f007:**
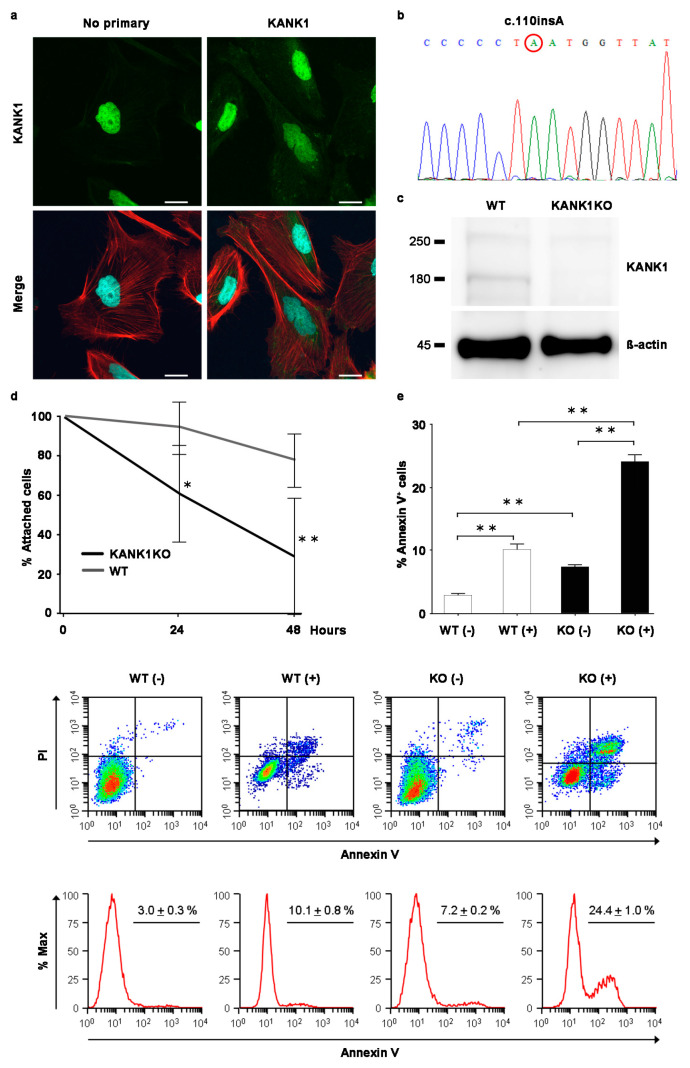
Generation of the human immortalized KANK1 knockout podocyte cell line. (**a**) The KANK1 protein was localized in the podocyte cytoplasm, co-localizing with the actin fibers’ edges as stained with rhodamine-phalloidin. Scale bars indicate 20 µm. (**b**) Sequence analysis identified a homozygous c.110ins A mutation in exon 2 of KANK1 in the KANK1KO podocytes. (**c**) The expression of the KANK1 protein was not detectable at approximately 180 kDa in KANK1KO podocytes, while there was a visible band in wild-type (WT) immortalized podocytes. The expression of β-actin was comparable between the two groups. (**d**) Attached cells were significantly decreased from 24 h after adriamycin treatment in KANK1KO podocytes compared to WT podocytes (94.7% ± 13.6% vs. 61.1% ± 24.6% after 24 h, * *p* < 0.05, 78.2% ± 13.8% vs. 29.2% ± 29.5% after 48 h, ** *p* < 0.01). (**e**) The percentage of apoptotic cells was significantly increased in KANK1KO podocytes compared to WT podocytes (** *p* < 0.01), and the percentage was increased significantly after adriamycin treatment (** *p* < 0.01).

## Data Availability

The data presented in this study are available on request from the corresponding author.

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
