# Peer review of "The Protective Role of KANK1 in Podocyte Injury"

_ijms, 2024, doi:10.3390/ijms25115808_

Round 1
Reviewer 1 Report
Comments and Suggestions for Authors
The authors with their work "The Protective Role of KANK1 in Podocyte Injury" investigate the role of KANK1 in nephrotic syndrome using both mouse models and human podocytes cell line. In the specific the goal of the research is to clarify KANK1's function in nephrotic syndrome. A systematic way to investigating this research topic is provided by the technique, which involves the creation of podocyte-specific Kank1 knockout mice and subsequent assessment of phenotypic alterations. The work is well structured, it successfully combines data from human podocytes and mouse models to provide a thorough knowledge of KANK1's function in a translational approach enanching the findings translational significance. An excellent strong point is that the results imply that KANK1 may protect podocytes from harm in pathological circumstances like nephrotic syndrome, which might have therapeutic ramifications. This emphasizes how crucial it is to look at KANK1 more as a possible therapeutic target for renal disorders. Beyond the useful information of the work, it is important to recognize its possible drawbacks, for example the requirement for more research to confirm the results in different models and clinical contexts.
Further knowledge of KANK1's therapeutic potential would benefit from establishing the specific molecular pathways underpinning its protective effect.
By the way the work can be implemented:
-Please, it is necessary to better introduce the pathology of nephrotic syndrome and also mention the genetic basis of this pathology, in the specific provide an overview about KANK gene family role in renal disease;
-It would have been very useful analyze the gene expression in Kank1 knockout mice and compare it to controls both in the healthy state and during renal damage. This might clarify KANK1's functional involvement in podocyte biology and assist discover downstream pathways or genes impacted by KANK1 loss;
-Immortalized cell lines frequently acquire genetic modifications during the immortalization process, resulting in genetic differences compared to primordial cells and it is possible that some pathways are altered. These changes can influence cellular function and may not accurately represent the physiology of basic cells. Why were primary cell lines not used in the experiments?
-In a patient, what improvements at a clinical/therapeutic level can knowledge of the damage caused by a mutation in KANK1 bring? Briefly mention it in the discussion;
Comments on the Quality of English Language
Moderate editing of English language required
Author Response
To Reviewer 1
The authors with their work "The Protective Role of KANK1 in Podocyte Injury" investigate the role of KANK1 in nephrotic syndrome using both mouse models and human podocytes cell line. In the specific the goal of the research is to clarify KANK1's function in nephrotic syndrome. A systematic way to investigating this research topic is provided by the technique, which involves the creation of podocyte-specific Kank1 knockout mice and subsequent assessment of phenotypic alterations. The work is well structured, it successfully combines data from human podocytes and mouse models to provide a thorough knowledge of KANK1's function in a translational approach enanching the findings translational significance. An excellent strong point is that the results imply that KANK1 may protect podocytes from harm in pathological circumstances like nephrotic syndrome, which might have therapeutic ramifications. This emphasizes how crucial it is to look at KANK1 more as a possible therapeutic target for renal disorders. Beyond the useful information of the work, it is important to recognize its possible drawbacks, for example the requirement for more research to confirm the results in different models and clinical contexts.
Further knowledge of KANK1's therapeutic potential would benefit from establishing the specific molecular pathways underpinning its protective effect.
By the way the work can be implemented:
-Please, it is necessary to better introduce the pathology of nephrotic syndrome and also mention the genetic basis of this pathology, in the specific provide an overview about KANK gene family role in renal disease;
Response
Thank you for this suggestion. As suggested, we have added the following passages to the Introduction section:
“The proteins encoded by 27 genes act as functional proteins (e.g., glomerular slit membrane components, laminin/integrin signaling components, actin-binding proteins, actin-regulating small GTPases, lysosomal proteins, transcription factors and proteins of coenzyme Q10 biosynthesis) [5].” (page 1, line 36-39)
“KANK family proteins play a role in actin regulation, which might be similar to ARHGDIA, the mutation of which is reported to cause SRNS [5].” (page 1, line 42-44).
-It would have been very useful analyze the gene expression in Kank1 knockout mice and compare it to controls both in the healthy state and during renal damage. This might clarify KANK1's functional involvement in podocyte biology and assist discover downstream pathways or genes impacted by KANK1 loss;
Response
Thank you for this suggestion. Unfortunately, we did not analyze the gene expression in Kank1-knockout mice or control mice in a healthy state or during renal damage. We added the following sentence to the Discussion:
"Second, gene expression analyses were not performed in Kank1-knockout mice and control mice in a healthy state or during renal damage." (page 12, line 261-262).
-Immortalized cell lines frequently acquire genetic modifications during the immortalization process, resulting in genetic differences compared to primordial cells and it is possible that some pathways are altered. These changes can influence cellular function and may not accurately represent the physiology of basic cells. Why were primary cell lines not used in the experiments?
Response
Thank you for this suggestion. Primary glomerular culture could not be performed because of technical problems. We have added this limitation to the Discussion section (page 12, line 268-269).
-In a patient, what improvements at a clinical/therapeutic level can knowledge of the damage caused by a mutation in KANK1 bring? Briefly mention it in the discussion;
Response
Thank you for this suggestion. We have added the following passage to the Discussion section:
“Therapeutic strategies for KANK1 gene mutations have not yet been established, and further research is required. As it is predicted that those with KANK1 gene mutations are more likely to develop kidney diseases, it is important for people with KANK1 gene mutations to be aware of the risk of developing kidney diseases and to strive to protect their kidneys.” (page 13, line 276-280).

Reviewer 2 Report
Comments and Suggestions for Authors
This time, Oda et al. focused on KANK, which is thought to be one of the causative genes of intractable nephrotic syndrome. They created KANK knockout model animals and evaluated their nephrotic status. There was no difference in urine protein in the KANK knockout model mice, but in a state of adriamycin-induced kidney damage, KANK-deficient mice had more protein in their urine.
The most difficult and important aspect of this research is the creation of a KANK knockout model animal, but based on the paper, it appears that the creation of the animal model was successful.
1. I don't think it would be strange if KANK knockout mice had a lot of protein in their urine even when they were not treated with adriamycin. I thought it would be a good idea to discuss in more detail the areas where this result did not result. Also, even if urinary protein was not found, what is the possibility that there is a difference in other urinary biomarkers? Please consider mentioning things like limitation.
2. I do not think that the adriamycin-induced renal failure state necessarily reflects the state of nephrosis, but have you considered other renal failure induction models? If you have not done so, please write it in the limitations section.
3. If you have any data, please describe whether the effects of drugs used for nephrotic syndrome, such as ARB, ACE-I, SGLT2 inhibitors, and steroids, are different in KANKknockout mice.
Comments on the Quality of English LanguageN/A
Author Response
To Reviewer 2
This time, Oda et al. focused on KANK, which is thought to be one of the causative genes of intractable nephrotic syndrome. They created KANK knockout model animals and evaluated their nephrotic status. There was no difference in urine protein in the KANK knockout model mice, but in a state of adriamycin-induced kidney damage, KANK-deficient mice had more protein in their urine.
The most difficult and important aspect of this research is the creation of a KANK knockout model animal, but based on the paper, it appears that the creation of the animal model was successful.
- I don't think it would be strange if KANK knockout mice had a lot of protein in their urine even when they were not treated with adriamycin. I thought it would be a good idea to discuss in more detail the areas where this result did not result. Also, even if urinary protein was not found, what is the possibility that there is a difference in other urinary biomarkers? Please consider mentioning things like limitation.
Response
Thank you for this suggestion. We added the following passages to the Discussion section: “Compensation by other members of the Kank family was not evident in the present study.” (page 12, line 231-232).
We did not test urinary biomarkers other than those reflecting the renal function and urinary protein, which was a limitation of this study. We have added this limitation to the Discussion section (page 12, line 262-263).
- I do not think that the adriamycin-induced renal failure state necessarily reflects the state of nephrosis, but have you considered other renal failure induction models? If you have not done so, please write it in the limitations section.
Response
Thank you for this suggestion. We did not use any other renal failure induction models in this study. We have added this limitation of to the Discussion section (page 12, line 259-261).
- If you have any data, please describe whether the effects of drugs used for nephrotic syndrome, such as ARB, ACE-I, SGLT2 inhibitors, and steroids, are different in KANKknockout mice.
Response
Thank you for this suggestion. We did not have data on drug effects in this experiment. This is one of the limitations of our study. We have added this limitation to the Discussion section (page 12, line 263-266).

Round 2
Reviewer 1 Report
Comments and Suggestions for Authors
Tha authors performed the modifications required.
Comments on the Quality of English LanguageModerate editing of English language required